# Vision-Based Defect Inspection and Condition Assessment for Sewer Pipes: A Comprehensive Survey

**DOI:** 10.3390/s22072722

**Published:** 2022-04-01

**Authors:** Yanfen Li, Hanxiang Wang, L. Minh Dang, Hyoung-Kyu Song, Hyeonjoon Moon

**Affiliations:** 1Department of Computer Science and Engineering, Sejong University, Seoul 05006, Korea; 1826535091@sju.ac.kr (Y.L.); hanxiang@sju.ac.kr (H.W.); 2Department of Information and Communication Engineering and Convergence Engineering for Intelligent Drone, Sejong University, Seoul 05006, Korea; minhdl@sejong.ac.kr (L.M.D.); songhk@sejong.ac.kr (H.-K.S.)

**Keywords:** survey, computer vision, defect inspection, condition assessment, sewer pipes

## Abstract

Due to the advantages of economics, safety, and efficiency, vision-based analysis techniques have recently gained conspicuous advancements, enabling them to be extensively applied for autonomous constructions. Although numerous studies regarding the defect inspection and condition assessment in underground sewer pipelines have presently emerged, we still lack a thorough and comprehensive survey of the latest developments. This survey presents a systematical taxonomy of diverse sewer inspection algorithms, which are sorted into three categories that include defect classification, defect detection, and defect segmentation. After reviewing the related sewer defect inspection studies for the past 22 years, the main research trends are organized and discussed in detail according to the proposed technical taxonomy. In addition, different datasets and the evaluation metrics used in the cited literature are described and explained. Furthermore, the performances of the state-of-the-art methods are reported from the aspects of processing accuracy and speed.

## 1. Introduction

### 1.1. Background

Underground sewerage systems (USSs) are a vital part of public infrastructure that contributes to collecting wastewater or stormwater from various sources and conveying it to storage tanks or sewer treatment facilities. A healthy USS with proper functionality can effectively prevent urban waterlogging and play a positive role in the sustainable development of water resources. However, sewer defects caused by different influence factors such as age and material directly affect the degradation of pipeline conditions. It was reported in previous studies that the conditions of USSs in some places are unsatisfactory and deteriorate over time. For example, a considerable proportion (20.8%) of Canadian sewers is graded as poor and very poor. The rehabilitation of these USSs is needed in the following decade in order to ensure normal operations and services on a continuing basis [1]. Currently, the maintenance and management of USSs have become challenging problems for municipalities worldwide due to the huge economic costs [2]. In 2019, a report in the United States of America (USA) estimated that utilities spent more than USD 3 billion on wastewater pipe replacements and repairs, which addressed 4692 miles of pipeline [3].

### 1.2. Defect Inspection Framework

Since it was first introduced in the 1960s [4], computer vision (CV) has become a mature technology that is used to realize promising automation for sewer inspections. In order to meet the increasing demands on USSs, a CV-based defect inspection system is required to identify, locate, or segment the varied defects prior to the rehabilitation process. As illustrated in Figure 1, an efficient defect inspection framework for underground sewer pipelines should cover five stages. In the data acquisition stage, there are many available techniques such as closed-circuit television (CCTV), sewer scanner and evaluation technology (SSET), and totally integrated sonar and camera systems (TISCITs) [5]. CCTV-based inspections rely on a remotely controlled tractor or robot with a mounted CCTV camera [6]. An SSET is a type of method that acquires the scanned data from a suite of sensor devices [7]. The TISCIT system utilizes sonar and CCTV cameras to obtain a 360° view of the sewer conditions [5]. As mentioned in several studies [6,8,9,10], CCTV-based inspections are the most widely used methods due to the advantages of economics, safety, and simplicity. Nevertheless, the performance of CCTV-based inspections is limited by the quality of the acquired data. Therefore, image-based learning methods require pre-processing algorithms to remove noise and enhance the resolution of the collected images. Many studies on sewer inspections have recently applied image pre-processing before examining the defects [11,12,13].

In the past few decades, many defect inspection strategies and algorithms have been presented based on CCTV cameras. Manual inspections by humans are inefficient and error-prone, so several studies attempted to adopt conventional machine learning (ML) approaches in order to diagnose the defects based on morphological, geometrical, or textural features [14,15,16]. With the elevation and progress of ML, deep learning (DL) methods have been widely applied to enhance the overall performance in recent studies on sewer inspections. Previous investigations have reviewed and summarized different kinds of inspections, which mainly include manual inspections [17,18] and automatic inspections based on the conventional machine learning algorithms [15,19] and deep learning algorithms [9,20].

In the attempt to evaluate the infrastructure conditions, some researchers have developed risk assessment approaches using different image post-processing algorithms [21,22,23]. For instance, a defect segmentation method was proposed to separate the cracks from the background, and post-processing was subsequently used to calculate the morphological features of the cracks [22]. In another study, a method based on a fully convolutional network and post-processing was introduced to detect and measure cracks [21]. Nevertheless, the existing risk assessment methods are limited to the feature analysis of cracks only, and there is no further research and exploration of each specific category.

### 1.3. Previous Survey Papers

Table 1 lists the major contributions of five survey papers, which considered different aspects of defect inspection and condition assessment in underground sewer pipelines. In 2019, an in-depth survey was presented to analyze different inspection algorithms [24]. However, it only focused on defect detection, and defect segmentation was not involved in this study. Several surveys [7,10,20] were conducted one year later to discuss the previous studies on sewer defects. Moreover, the recent studies associated with image-based construction applications are discussed in [8]. In these relevant surveys, the authors of each paper put efforts into emphasizing a particular area. A more comprehensive review of the latest research on defect inspection and condition assessment is significant for the researchers who are interested in integrating the algorithms into real-life sewer applications. In addition, the detailed and well-arranged list tables for the existing defect inspection methods according to the different categories are not provided in these papers.

### 1.4. Contributions

In order to address the above issues, a survey that covers various methods regarding sewer defect inspection and condition assessment is conducted in this study. The main contributions are as follows. This survey creates a comprehensive review of the vision-based algorithms about defect inspection and condition assessment from 2000 to the present. Moreover, we divide the existing algorithms into three categories, which include defect classification, detection, and segmentation. In addition, different datasets and evaluation metrics are summarized. Based on the investigated papers, the research focuses and tendencies in previous studies are analyzed. Meanwhile, the limitations of the existing approaches as well as the future research directions are indicated.

The rest of this survey is divided into four sections. Section 2 presents the methodology used in this survey. Section 3 discusses the image-based defect inspection algorithms that cover classification, detection, and segmentation. Section 4 analyzes the dataset and the evaluation metrics that are used from 2000 onwards. In Section 5, the challenges and future needs are indicated. Conclusions of previous studies and suggestions for future research are provided in Section 6.

## 2. Survey Methodology

A thorough search of the academic studies was conducted by using the Scopus journal database. It automatically arranges the results from different publishers, which include Elsevier, Springer Link, Wiley online library, IEEE Xplore, ASCE Library, MDPI, SACG, preprint, Taylor & Francis Group, and others. Figure 2 shows the distribution of the academic journals reviewed in diverse databases. The journals in the other databases include SPIE Digital Library, Korean Science, Easy Chair, and Nature. In order to highlight the advances in vision-based defect inspection and condition assessment, the papers of these fields that were published between 2000 and 2022 are investigated. The search criterion of this survey is to use an advanced retrieval approach by selecting high-level keywords like (“vision-based sensor” OR “video” OR “image”) AND (“automatic sewer inspection” OR “defect classification” OR “defect detection” OR “defect segmentation” OR “condition assessment”). Since there is no limitation on a certain specific construction material or pipe typology, the research on any sewer pipeline that can be entered and that obtained visual data is covered in this survey. Nevertheless, the papers that focus on some topics, which do not relate to the vision-based sewer inspection, are not included in this paper. For example, the quality assessment for sewer images [25], pipe reconstruction, internal pipe structure, wall thickness measurement, and sewer inspections based on other sensors such as depth sensors [26,27], laser scanners [28,29], or acoustic sensors [30,31] are considered irrelevant topics. Figure 3 represents the number of articles including journals and conference papers in different time periods from 2000 to 2022. By manually scanning the title and abstract sections, a total of 124 papers that includes both journals (95) and conferences (29) in English was selected to examine the topic’s relevancy. In addition to these papers, four books and three websites were also used to construct this survey. After that, the filtered papers were classified in terms of the employed methods and application areas. Finally, the papers in each category were further studied by analyzing their weaknesses and strengths.

## 3. Defect Inspection

In this section, several classic algorithms are illustrated, and the research tendency is analyzed. Figure 4 provides a brief description of the algorithms in each category. According to the literature review, the existing studies about sewer inspection are summarized in three tables. Table 2, Table 3 and Table 4 show the recent studies about defect classification (Section 3.1), detection (Section 3.2), and segmentation (Section 3.3) algorithms. In order to comprehensively analyze these studies, the publication time, title, utilized methodology, advantages, and disadvantages for each study are covered. Moreover, the specific proportion of each inspection algorithm is computed in Figure 5. It is clear that the defect classification accounts for the most significant percentages in all the investigated studies.

### 3.1. Defect Classification

Due to the recent advancements in ML, both the scientific community and industry have attempted to apply ML-based pattern recognition in various areas, such as agriculture [32], resource management [33], and construction [34]. At present, many types of defect classification algorithms have been presented for both binary and multi-class classification tasks. The commonly used algorithms are described below.

#### 3.1.1. Support Vector Machines (SVMs)

SVMs have become one of the most typical and robust ML algorithms because they are not sensitive to the overfitting problem compared with other ML algorithms [35,36,37]. The principal objective of an SVM is to perfectly divide the training data into two or more classes by optimizing the classification hyperplane [38,39]. A classification hyperplane equation can be normalized in order to form a two-dimensional sample set that satisfies Equation (1).
(1)yi(wTx+b)≥1, i=1, …,n.
where xi∈ℝ2 and yi∈(+1,−1); w is the optimal separator and b is the bias. As shown in Figure 6, the circles and triangles indicate two classes of training samples. The optimal hyperplane is represented as H, and the other two parallel hyperplanes are represented as H_1_ and H_2_. On the premise of correctly separating samples, the maximum margin between the two hyperplanes (H_1_ and H_2_) is conducive to gaining the optimal hyperplane (H).

Despite classifying various types of defects with high accuracy, the SVM algorithm cannot be applied to end-to-end classification problems [40]. As demonstrated in [41], Ye et al. established a sewer image diagnosis system where a variety of image pre-processing algorithms, such as Hu invariant moments [42] and lateral Fourier transform [43] were used for the feature extraction, and the SVM was then used as the classifier. The accuracy of the SVM classifier reached 84.1% for seven predefined classes, and the results suggested that the training sample number is positively correlated with the final accuracy. In addition to this study, Zuo et al. applied the SVM algorithm that is based on a specific histogram to categorize three different cracks at the sub-class level [11]. Before the classification process, bilateral filtering [44,45] was applied in image pre-processing in order to denoise input images and keep the edge information. Their proposed method obtained a satisfactory average accuracy of 89.6%, whereas it requires a series of algorithms to acquire 2D radius angular features before classifying the defects.

#### 3.1.2. Convolutional Neural Networks (CNNs)

A CNN was first proposed in 1962 [46], and it has demonstrated excellent performances in multiple domains. Due to its powerful generalization ability, CNN-based classifiers that automatically extract features from input images are superior to the classifiers that are based on the pre-engineered features [47]. Consequently, numerous researchers have applied CNNs to handle the defect classification problem in recent years. Kumar et al. presented an end-to-end classification method using several binary CNNs in order to identify the presence of three types of commonly encountered defects in sewer images [48]. In their proposed framework, the extracted frames were inputted into networks that contained two convolutional layers, two pooling layers, two fully connected layers, and one output layer. The classification results achieved high values in terms of average accuracy (0.862), precision (0.877), and recall (0.906), but this work was limited to the classification of ubiquitous defects. 

Meijer et al. reimplemented the network proposed in [48], and they compared the performances based on a more realistic dataset introduced in [49]. They used a single CNN to deal with multi-label classification problems, and their classifier outperformed the method presented by Kumar et al. In another work, several image pre-processing approaches, which included histogram equalization [50] and morphology operations [51], were used for noise removal. After that, a fine-tuned defect classification model was used to extract informative features based on highly imbalanced data [52]. Their presented model architecture was based on the VGG network, which achieved first place in the ILSVRC-2014 [53]. As illustrated in Figure 7, the model structure in the first 17 layers is frozen, and the other sections are trainable; also, two convolutional layers and one batch normalization were added to enhance the robustness of the modified network.

**Table 2 sensors-22-02722-t002:** Academic studies in vision-based defect classification algorithms.

Time	Methodology	Advantage	Disadvantage	Ref.
2000	Back-propagation algorithm	Perform well for classification	Slow learning speed	[54]
2002	Neuro-fuzzy algorithm	Good classification efficiency	Weak feature extraction scheme	[55]
2006	Neuro-fuzzy classifier	● Combines neural network and fuzzy logic concepts ● Screens data before network training to improve efficiency	Not an end-to-end model	[56]
2009	Rule-based classifier	Recognize defects under the realistic sewer condition	No real-time recognition	[57]
2009	Rule-based classifier	Addresses realistic defect detection and recognition	Unsatisfactory classification result	[58]
2009	Radial basis network (RBN)	Overall classification accuracy is high	Heavily relies on the pre-engineered results	[59]
2012	Self-organizing map (SOM)	Suitable for large-scale real applications	High computation complexities	[60]
2013	Ensemble classifiers	● High practicability ● Reliable classification result	Feature extraction and classification are separately implemented	[61]
2016	Random forests	Dramatically reduces the processing time	Processing speed can be improved	[62]
2017	Random forest classifier	Automated fault classification	Poor performance	[63]
2017	Hidden Markov model (HMM)	● Efficient for numerous patterns of defects ● Real time	Low classification accuracy	[64]
2018	One-class SVM (OCSVM)	Available for both still images and video sequences	Cannot achieve a standard performance	[65]
2018	Multi-class random forest	Poor classification accuracy	Real-time prediction	[66]
2018	Multiple binary CNNs	● Good generalization capability ● Can be easily re-trained	● Do not support sub-defects classification ● Cannot localize defects in pipeline	[48]
2018	CNN	● High detection accuracy ● Strong scene adaptability in realistic scenes	Poor performance for the unnoticeable defects	[67]
2018	HMM and CNN	Automatic defect detection and classification in videos	Poor performance	[68]
2019	Single CNN	● Outperforms the SOTA ● Allows multi-label classification	Weak performance for fully automatic classification	[49]
2019	Two-level hierarchical CNNs	Can identify the sewer images into different classes	Cannot classify multiple defects in the same image simultaneously	[69]
2019	Deep CNN	● Classifies defects at different levels ● Performs well in classifying most classes	There exists a extremely imbalanced data problem (IDP)	[70]
2019	CNN	Accurate recognition and localization for each defect	Classifies only one defect with the highest probability in an image	[71]
2019	SVM	Reveals the relationship between training data and accuracy	Requires various steps for feature extraction	[41]
2020	SVM	● Classifies cracks at a sub-category level ● High recall and fast processing speed	Limited to only three crack patterns	[11]
2020	CNN	● Image pre-processing used for noisy removal and image enhancement ● High classification accuracy	Can classify one defect only	[72]
2020	CNN	Shows great ability for defect classification under various conditions	Limited to recognize the tiny and narrow cracks	[73]
2021	CNN	Is robust against the IDP and noisy factors in sewer images	No multi-label classification	[52]
2021	CNN	Covers defect classification, detection, and segmentation	Weak classification results	[74]

### 3.2. Defect Detection

Rather than the classification algorithms that merely offer each defect a class type, object detection is conducted to locate and classify the objects among the predefined classes using rectangular bounding boxes (BBs) as well as confidence scores (CSs). In recent studies, object detection technology has been increasingly applied in several fields, such as intelligent transportation [75,76,77], smart agriculture [78,79,80], and autonomous construction [81,82,83]. The generic object detection consists of the one-stage approaches and the two-stage approaches. The classic one-stage detectors based on regression include YOLO [84], SSD [85], CornerNet [86], and RetinaNet [87]. The two-stage detectors are based on region proposals, including Fast R-CNN [88], Faster R-CNN [89], and R-FCN [90]. In this survey, the one-stage and two-stage methods that were employed in sewer inspection studies are both discussed and analyzed as follows.

#### 3.2.1. You Only Look Once (YOLO)

YOLO is a one-stage algorithm that maps directly from image pixels to BBs and class probabilities. In [84], object detection was addressed as a single regression problem using a simple and unified pipeline. Due to its advantages of robustness and efficiency, an updated version of YOLO, which is called YOLOv3 [91], was explored to locate and classify defects in [9]. YOLOv3 outperformed the previous YOLO algorithms in regard to detecting the objects with small sizes because the YOLOv3 model applies a 3-scale mechanism that concatenates the feature maps of three scales [92,93]. Figure 8 illustrates how the YOLOv3 architecture implements the 3-scale prediction operation. The prediction result with a scale of 13 × 13 is obtained in the 82nd layer by down-sampling and convolution operations. Then, the result in the 79th layer is concatenated with the result of the 61st layer after up-sampling, and the prediction result with 26 × 26 is generated after several convolution operations. The result of 52 × 52 is generated at layer 106 using the same method. The predictions at different scales have different receptive fields that determine the appropriate sizes of the detection objects in the image. As a result, YOLOv3 with a 3-scale mechanism is capable of detecting more fine-grained features.

Based on the detection model developed by [9], a video interpretation algorithm was proposed to build an autonomous assessment framework in sewer pipelines [94]. The assessment system verified how the defect detector can be put to use with realistic infrastructure maintenance and management. A total of 3664 images extracted from 63 videos were trained by the YOLOv3 model, which achieved a high mean average precision (mAP) of 85.37% for seven defects and also obtained a fast detection speed for real-time applications.

#### 3.2.2. Single Shot Multibox Detector (SSD)

Similarly, another end-to-end detector that is named SSD was first introduced for multiple object classes in [85]. Several experiments were conducted to analyze the detection speed and accuracy based on different public datasets. The results suggest that the SSD model (input size: 300 × 300) obtained faster speed and higher accuracy than the YOLO model (input size: 448 × 448) on the VOC2007 test. As shown in Figure 9, the SSD method first extracts features in the base network (VGG16 [53]). It then predicts the fixed-size bounding boxes and class scores for each object instance using a feed-forward CNN [95]. After that, a non-maximum suppression (NMS) algorithm [96] is used to refine the detections by removing the redundant boxes.

Moreover, the SSD method was utilized to detect defects for CCTV images in a condition assessment framework [97]. Several image pre-processing algorithms were used to enhance the input images prior to the feature extraction process. Then three state-of-the-art (SOTA) detectors (YOLOv3 [91], SSD [85], and faster-RCNN [89]) based on DLs were tested and compared on the same dataset. The defect severity was rated in the end from different aspects in order to assess the pipe condition. Among three experimental models, YOLOv3 demonstrated that it obtained a relatively balanced performance between speed and accuracy. The SSD model achieved the fastest speed (33 ms per image), indicating the feasibility of real-time defect detection. However, the detection accuracy of SSD was the lowest, which was 28.6% lower than the accuracy of faster R-CNN.

#### 3.2.3. Faster Region-Based CNN (Faster R-CNN)

The faster R-CNN model was introduced to first produce candidate BBs and then refine the generated BB proposals [89]. Figure 10 shows the architecture of faster R-CNN developed by [98] in a defect detection system. First of all, the multiple CNN layers in the base network were used for feature extraction. Then, the region proposal network (RPN) created numerous proposals based on the generated feature maps. Finally, these proposals were sent to the detector for further classification and localization. Compared with the one-stage frameworks, the region proposal-based methods require more time in handling different model components. However, the faster R-CNN model that trains RPN and fast R-CNN detector separately is more accurate than other end-to-end training models, such as YOLO and SSD [99]. As a result, the faster R-CNN was explored in many studies for more precise detection of sewer defects.

In [98], 3000 CCTV images were fed into the faster R-CNN model, and the trained model was then utilized to detect four categories of defects. This research indicated that the data size, training scheme, network structure, and hyper-parameter are important impact factors for the final detection accuracy. The results show the modified model achieved a high mAP of 83%, which was 3.2% higher than the original model. In another work [99], a defect tracking framework was firstly built by using a faster R-CNN detector and learning discriminative features. In the defect detection process, a mAP of 77% was obtained for detecting three defects. At the same time, the metric learning model was trained to reidentify defects. Finally, the defects in CCTV videos were tracked based on detection information and learned features.

**Table 3 sensors-22-02722-t003:** Academic studies in vision-based defect detection algorithms.

Time	Methodology	Advantage	Disadvantage	Ref.
2004	Genetic algorithm (GA) and CNN	High average detection rate	Can only detect one type of defect	[100]
2014	Histograms of oriented gradients (HOG) and SVM	Viable and robust algorithm	Complicated image processing steps before detecting defective regions	[101]
2018	Faster R-CNN	● Explores the influences of several factors for the model performance ● Provides references to applied DL in autonomous construction	Limited to the still images	[98]
2018	Faster R-CNN	Addresses similar object detection problems in industry	Long training time and slow detection speed	[102]
2018	Rule-based detection algorithm	● Based on image processing techniques ● No need training process ● Requires less images	Low detection performance	[103]
2019	YOLO	End-to-end detection workflow	Cannot detect defect at the sub-classes	[104]
2019	YOLOv3	● High detection rate ● Real-time defect detection ● Efficient input data manipulation process	Weak function of output frames	[9]
2019	SSD, YOLOv3, and Faster R-CNN	Automatic detection for the operational defects	Cannot detect the structural defects	[105]
2019	Rule-based detection algorithm	Performs well on the low-resolution images	Requires multiple digital image processing steps	[106]
2019	Kernel-based detector	Promising and reliable results for anomaly detection	Cannot get the true position inside pipelines	[107]
2019	CNN and YOLO	Obtained a considerable reduction in processing speed	Can detect only one type of structural defect	[108]
2020	Faster R-CNN	Can assess the defect severity as well as the pipe condition	Cannot run in real time	[97]
2020	Faster R-CNN	● Can obtain the number of defects ● First work for sewer defect tracking	Requires training two models separately, not an end-to-end framework	[99]
2020	SSD, YOLOv3, and Faster R-CNN	Automated defect detection	Structural defect detection and severity classification are not available	[105]
2021	YOLOv3	● Covers defect detection, video interpretation, and text recognition ● Detect defect in real time	The ground truths (GTs) are not convincing	[94]
2021	CNN and non-overlapping windows	Outperformed existing models in terms of detection accuracy	Deeper CNN model with better performance requires longer inference time	[109]
2021	Strengthened region proposal network (SRPN)	● Effectively locate defects ● Accurately assess the defect grade	● Cannot be applied for online processing ● Cannot identify if the defect is mirrored	[110]
2021	YOLOv2	Covers defect classification, detection, and segmentation	Weak detection results	[74]
2022	Transformer-based defect detection (DefectTR)	● Does not require prior knowledge ● Can generalize well with limited parameters	The robustness and efficiency can be improved for real-world applications	[111]

### 3.3. Defect Segmentation

Defect segmentation algorithms can predict defect categories and pixel-level location information with exact shapes, which is becoming increasingly significant for the research on sewer condition assessment by re-coding the exact defect attributes and analyzing the specific severity of each defect. The previous segmentation methods were mainly based on mathematical morphology [112,113]. However, the morphology segmentation approaches were inefficient compared to the DL-based segmentation methods. As a result, the defect segmentation methods based on DL have been recently explored in various fields. The studies related to sewer inspection are described as follows.

#### 3.3.1. Morphology Segmentation

Morphology-based defect segmentation contains many methods, such as closing bottom-hat operation (CBHO), opening top-hat operation (OTHO), and morphological segmentation based on edge detection (MSED). By evaluating and comparing the segmentation performances of different methods, MSED was verified as being useful to detect cracks, and OTHO was verified as being useful to detect open joints [113]. They also indicated the removal of the text on the CCTV images is necessary to further improve the detection accuracy. Similarly, MSED was applied to segment eight categories of typical defects, and it outperformed another popular approach called OTHO [112]. In addition, some morphology features, including area, axis length, and eccentricity, were also measured, which is of great significance to assist inspectors in judging and assessing defect severity. Although the morphology segmentation methods showed good segment results, they need multiple image pre-processing steps before the segmentation process.

#### 3.3.2. Semantic Segmentation

Automatic localization of the sewer defect’s shape and the boundary was first proposed by Wang et al. using a semantic segmentation network called DilaSeg [114]. In order to improve the segmentation accuracy, an updated network named DilaSeg-CRF was introduced by combining the CNN with a dense conditional random field (CRF) [115,116]. Their updated network improved the segmentation accuracy considerably in terms of the mean intersection over union (mIoU), but the single data feature and the complicated training process reflect that the DilaSe-CRF is not suitable to be applied in real-life applications. 

Recently, the fully convolutional network (FCN) has been explored for the pixels-to-pixels segmentation task [117,118,119,120]. Meanwhile, some other network architectures that are similar to an FCN have emerged in large numbers, including U-Net [121]. Pan et al. proposed a semantic segmentation network called PipeUNet, in which the U-Net was used as the backbone due to its fast convergence speed [122]. As shown in Figure 11, the encoder and decoder on both sides form a symmetrical architecture. In addition, three FRAM blocks were added before the skip connections to improve the ability of feature extraction and reuse. Besides, the focal loss was demonstrated, which is useful for handling the imbalanced data problem (IDP). Their proposed PipeUNet achieved a high mIoU of 76.3% and a fast speed of 32 frames per second (FPS).

**Table 4 sensors-22-02722-t004:** Academic studies in vision-based defect segmentation algorithms.

Time	Methodology	Advantage	Disadvantage	Ref.
2005	Mathematical morphology-based Segmentation	● Automated segmentation based on geometry image modeling ● Perform well under various environments	● Can only segment cracks ● Complicated and multiple steps	[123]
2014	Mathematical morphology-based Segmentation	Requires less data and computing resources to achieve a decent performance	● Challenging to detect cracks ● Various processing steps	[113]
2019	DL-based semantic segmentation (DilaSeg-CRF)	● End-to-end trainable model ● Fair inference speed	Long training time	[116]
2020	DilaSeg-CRF	● Promising segmentation accuracy ● The defect severity grade is presented	Complicated workflow	[23]
2020	DL-based semantic segmentation (PipeUNet)	● Enhances the feature extraction capability ● Resolves semantic feature differences ● Solves the IDP	Still exists negative segmentation results	[122]
2021	Feature pyramid networks (FPN) and CNN	Covers defect classification, detection, and segmentation	Weak segmentation results	[74]
2022	DL-based defect segmentation (Pipe-SOLO)	● Can segment defect at the instance level ● Is robust against various noises from natural scenes	Only suitable for still sewer images	[124]

## 4. Dataset and Evaluation Metric

The performances of all the algorithms were tested and are reported based on a specific dataset using specific metrics. As a result, datasets and protocols were two primary determining factors in the algorithm evaluation process. The evaluation results are not convincing if the dataset is not representative, or the used metric is poor. It is challenging to judge what method is the SOTA because the existing methods in sewer inspections utilize different datasets and protocols. Therefore, benchmark datasets and standard evaluation protocols are necessary to be provided for future studies.

### 4.1. Dataset

#### 4.1.1. Dataset Collection

Currently, many data collection robotic systems have emerged that are capable of assisting workers with sewer inspection and spot repair. Table 5 lists the latest advanced robots along with their respective information, including the robot’s name, company, pipe diameter, camera feature, country, and main strong points. Figure 12 introduces several representative robots that are widely utilized to acquire images or videos from underground infrastructures. As shown in Figure 12a, LETS 6.0 is a versatile and powerful inspection system that can be quickly set up to operate in 150 mm or larger pipes. A representative work (Robocam 6) of the Korean company TAP Electronics is shown in Figure 12b. Robocam 6 is the best model to increase the inspection performance without the considerable cost of replacing the equipment. Figure 12c is the X5-HS robot that was developed in China, which is a typical robotic crawler with a high-definition camera. In Figure 12d, Robocam 3000, sold by Japan, is the only large-scale system that is specially devised for inspecting pipes ranging from 250 mm to 3000 mm. It used to be unrealistic to apply the crawler in huge pipelines in Korea.

#### 4.1.2. Benchmarked Dataset

Open-source sewer defect data is necessary for academia to promote fair comparisons in automatic multi-defect classification tasks. In this survey, a publicly available benchmark dataset called Sewer-ML [125] for vision-based defect classification is introduced. The Sewer-ML dataset, acquired from Danish companies, contains 1.3 million images labeled by sewer experts with rich experience. Figure 13 shows some sample images from the Sewer-ML dataset, and each image includes one or more classes of defects. The recorded text in the image was redacted using a Gaussian blur kernel to protect private information. Besides, the detailed information of the datasets used in recent papers is described in Table 6. This paper summarizes 32 datasets from different countries in the world, of which the USA has 12 datasets, accounting for the largest proportion. The largest dataset contains 2,202,582 images, whereas the smallest dataset has only 32 images. Since the images were acquired by various types of equipment, the collected images have varied resolutions ranging from 64 × 64 to 4000 × 46,000.

### 4.2. Evaluation Metric

The studied performances are ambiguous and unreliable if there is no suitable metric. In order to present a comprehensive evaluation, multitudinous methods are proposed in recent studies. Detailed descriptions of different evaluation metrics are explained in Table 7. Table 8 presents the performances of the investigated algorithms on different datasets in terms of different metrics. 

As shown in Table 8, accuracy is the most commonly used metric in the classification tasks [41,48,52,54,56,57,58,61,65,66,67,69,70,71,73]. In addition to this, other subsidiary metrics such as precision [11,48,67,69,73], recall [11,48,57,67,69,73], and F1-score [69,73] are also well supported. Furthermore, AUROC and AUPR are calculated in [49] to measure the classification results, and FAR is used in [57,58] to check the false alarm rate in all the predictions. In contrast to classification, mAP is a principal metric for detection tasks [9,98,99,105,110]. In another study [97], precision, recall, and F1-score are reported in conjunction to provide a comprehensive estimation for defect detection. Heo et al. [106] assessed the model performance based on the detection rate and the error rate. Kumar and Abraham [108] report the average precision (AP), which is similar to the mAP but for each class. For the segmentation tasks, the mIoU is considered as an important metric that is used in many studies [116,122]. Apart from the mIoU, the per-class pixel accuracy (PA), mean pixel accuracy (mPA), and frequency-weighted IoU (fwIoU) are applied to evaluate the segmented results at the pixel level.

## 5. Challenges and Future Work

This part first discusses the main challenges in recent studies, and some potential methods are then indicated to address these difficulties in the future work. Since a few surveys have already mentioned the partial limitations, a more complete summary of the existing challenges and future research direction are presented in this survey.

### 5.1. Data Analysis

During the data acquisition process, vision-based techniques such as the traditional CCTV are the most popular because of their cost-effective characteristics. Nevertheless, it is challenging for the visual equipment to inspect all the defects whenever they are below the water level or behind pipelines. As a result, the progress in hybrid devices has provided a feasible approach to acquire unavailable defects [126]. For example, the SSET methods [31,127,128] have been applied to collect quality data and evaluate the detected defects that are hard to deduce based on the visual data. In addition, the existing sewer inspection studies mainly focus on the concrete pipeline structures. The inspection and assessment for the traditional masonry sewer system that are still ubiquitous in most of the European cities become cumbersome for inspectors in practice. As for this issue, several automated diagnostic techniques (CCTV, laser scanning, ultrasound, etc.) for brick sewers are analyzed and compared in detail by enumerating the specific advantages and disadvantages [129,130]. Furthermore, varied qualities of the exiting datasets under distinct conditions and discontinuous backgrounds require image preprocessing prior to the inspection process to enhance the image quality and then improve the final performance [131].

Moreover, the current work concentrates on the research of structural defects such as cracks, joints, breaks, surface damage, lateral protrusion, and deformation, whereas there is less concern about the operation and maintenance defects (roots, infiltration, deposits, debris, and barriers). As mentioned in Section 4.1.2, there are 32 datasets investigated in this survey. Figure 14 shows the previous studies on sewer inspections of different classes of defects. We listed 12 classes of common defects in underground sewer pipelines. In addition to this, other defects that are rare and at the sub-class level are also included. According to the statistics for common defects, the proportion (50.5%) of structural defects is 20.3% higher than the proportion (30.2%) of operation and maintenance defects, which reflects that future research needs more available data for operation and maintenance defects.

### 5.2. Model Analysis

Although defect severity analysis methods have been proposed in several papers in order to assess the risk of the detected cracks, approaches for the analysis of other defects are limited. As for cracks, the risk levels can be assessed by measuring morphological features such as the crack length, mean width, and area to judge the corresponding severity degree. In contrast, it is difficult to comprehensively analyze the severity degrees for other defects because only the defect area is available for other defects. Therefore, researchers should explore more features that are closely related to the defect severity, which is significant for further condition assessment.

In addition, the major defect inspection models rely on effective supervised learning methods that cost much time in the manual annotation process for training [10]. The completely automated systems that include automatic labeling tools need to be developed for more efficient sewer inspections. On the other hand, most of the inspection approaches that demand long processing times only test based on still images, so these methods cannot be practiced in real-time applications for live inspections. More efforts should be made in future research to boost the inference speed in CCTV sewer videos.

## 6. Conclusions

Vision-based automation in construction has attracted increasing interest of researchers from different fields, especially with image processing and pattern recognition. The main outcomes of this paper include (1) an exhaustive review of diverse research approaches presented in more than 120 studies through a scientific taxonomy, (2) an analytical discussion of various algorithms, datasets, and evaluation protocols, and (3) a compendious summary of the existing challenges and future needs. Based on the current research situation, this survey outlines several suggestions that can facilitate future research on vision-based sewer inspection and condition assessment. Firstly, classification and detection have become a topic of great interest in the past several decades, which has attracted a lot of researchers’ attention. Compared with them, defect segmentation at the pixel level is a more significant task to assist the sewer inspectors in evaluating the risk level of the detected defects. However, it has the lowest proportion of the research of overall studies. Hence, automatic defect segmentation should be given greater focus considering its research significance. Secondly, we suggest that a public dataset and source code be created to support replicable research in the future. Thirdly, the evaluation metrics should be standardized for a fair performance comparison. Since this review presents clear guidelines for subsequent research by analyzing the concurrent studies, we believe it is of value to readers and practitioners who are concerned with sewer defect inspection and condition assessment.

## Figures and Tables

**Figure 1 sensors-22-02722-f001:**
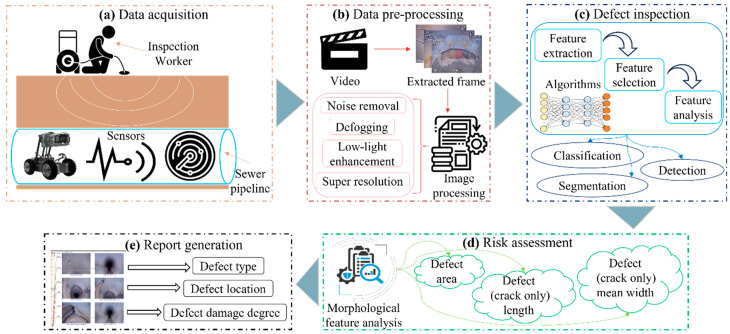
There are five stages in the defect inspection framework, which include (**a**) the data acquisition stage based on various sensors (CCTV, sonar, or scanner), (**b**) the data processing stage for the collected data, (**c**) the defect inspection stage containing different algorithms (defect classification, detection, and segmentation), (**d**) the risk assessment for detected defects using image post-processing, and (**e**) the final report generation stage for the condition evaluation.

**Figure 2 sensors-22-02722-f002:**
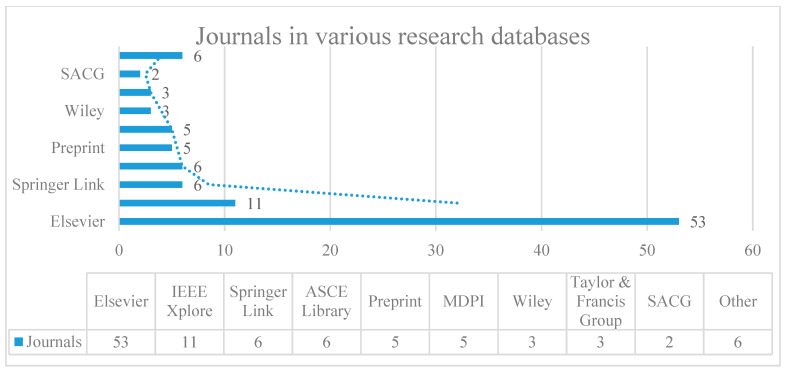
Number of journal publications investigated in different databases.

**Figure 3 sensors-22-02722-f003:**
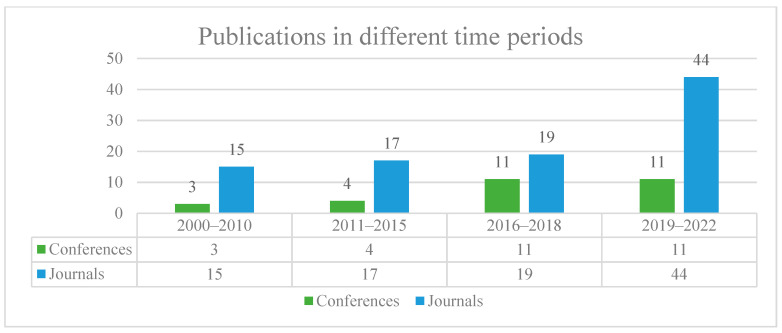
Number of publications investigated in different time periods (from 2000 to 2022).

**Figure 4 sensors-22-02722-f004:**
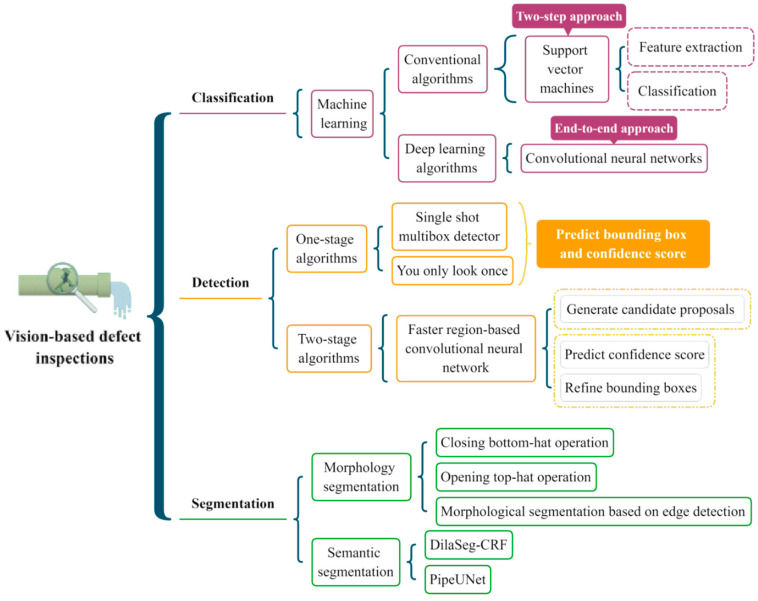
The classification map of the existing algorithms for each category. The dotted boxes represent the main stages of the algorithms.

**Figure 5 sensors-22-02722-f005:**
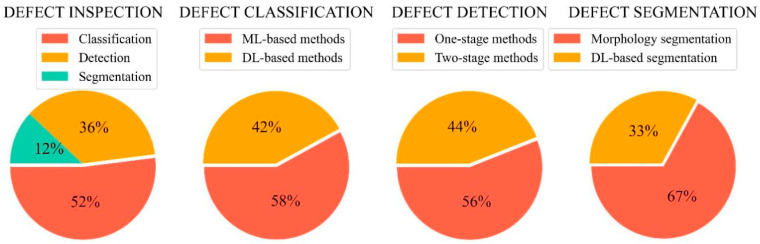
Proportions of the investigated studies using different inspection algorithms.

**Figure 6 sensors-22-02722-f006:**
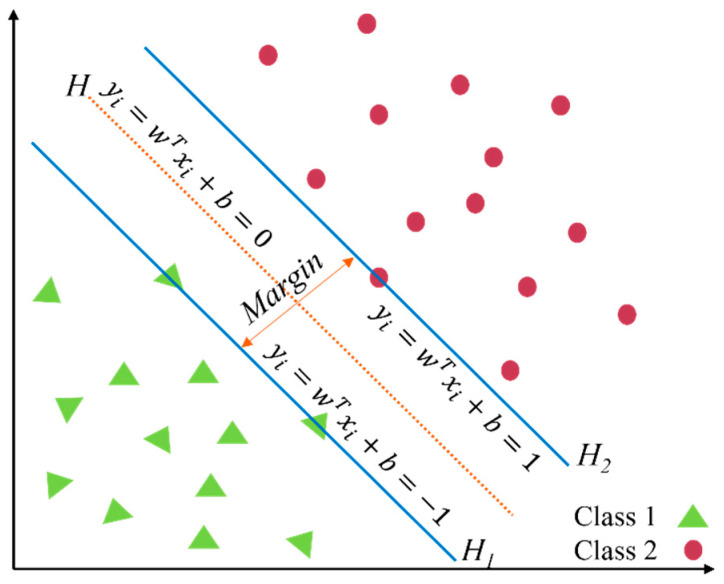
Optimal separation hyperplane.

**Figure 7 sensors-22-02722-f007:**
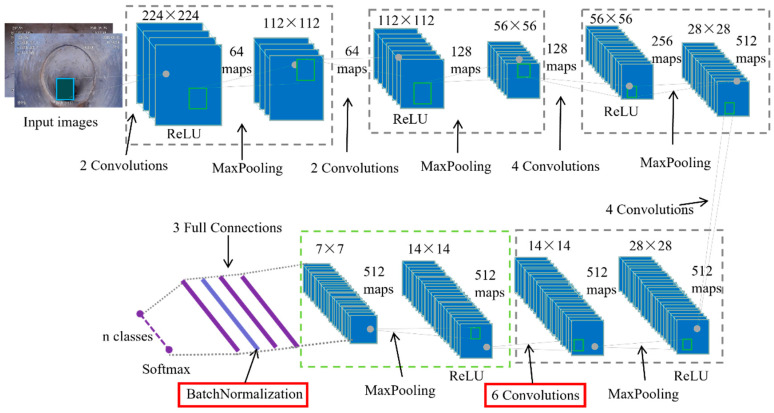
A fine-tuned network architecture used for defect classification.

**Figure 8 sensors-22-02722-f008:**
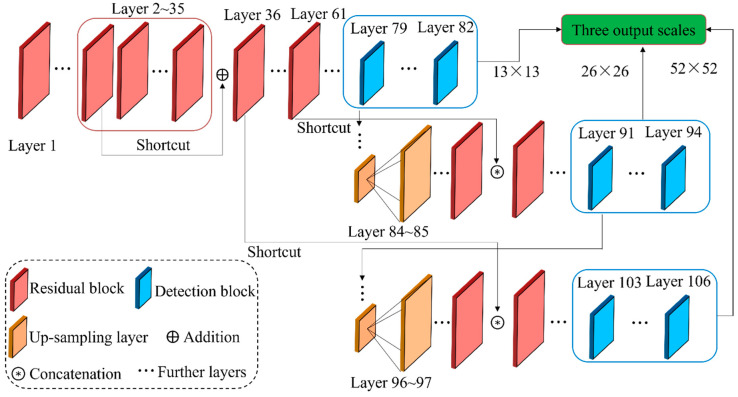
The YOLOv3 architecture with the 3-scale prediction mechanism.

**Figure 9 sensors-22-02722-f009:**
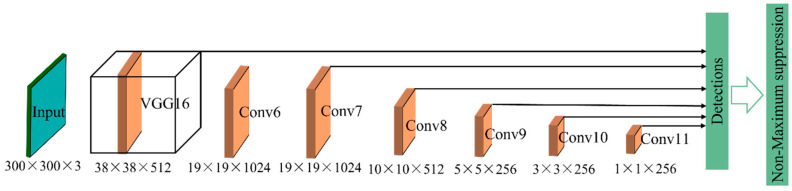
The model architecture of the SSD model.

**Figure 10 sensors-22-02722-f010:**
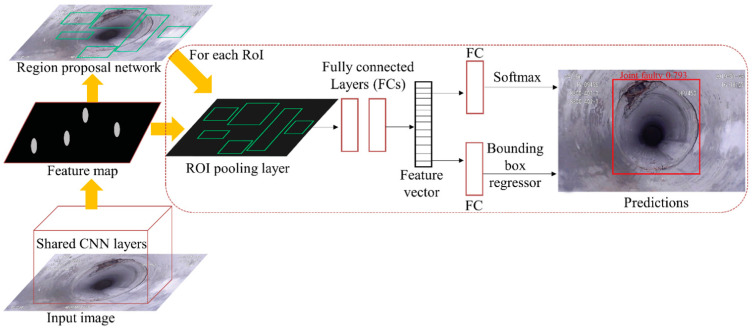
An architecture of the faster R-CNN developed for defect detection. ‘CNN’ refers to convolutional neural network. ‘ROI’ means region of interest. ‘FC’ is fully connected layer.

**Figure 11 sensors-22-02722-f011:**
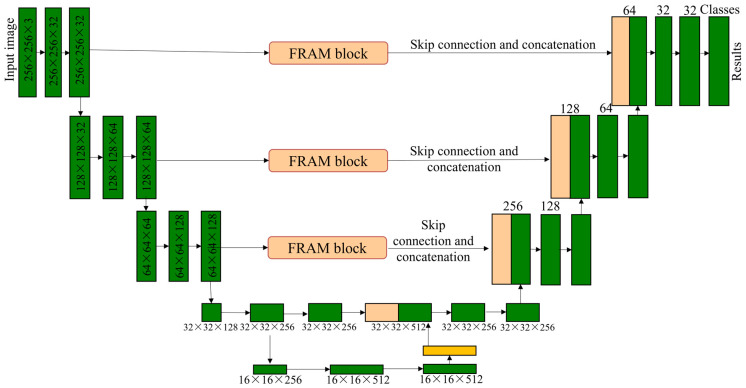
An architecture of PipeUNet proposed for semantic segmentation.

**Figure 12 sensors-22-02722-f012:**
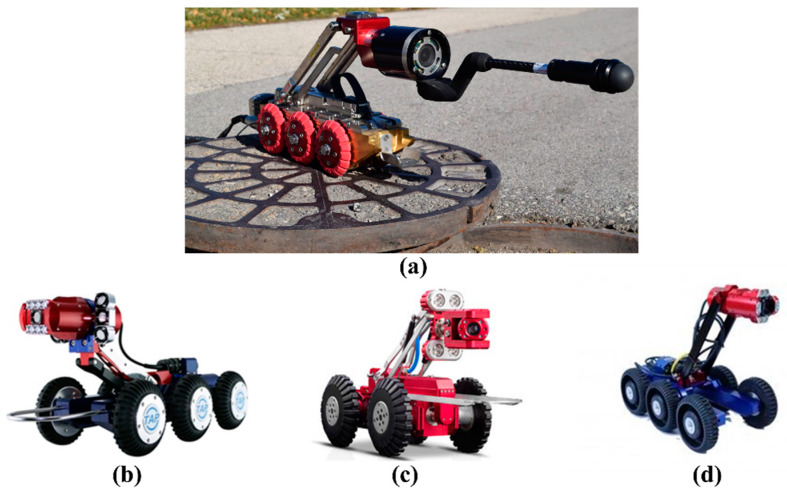
Representative inspection robots for data acquisition. (**a**) LETS 6.0, (**b**) Robocam 6, (**c**) X5-HS, and (**d**) Robocam 3000.

**Figure 13 sensors-22-02722-f013:**
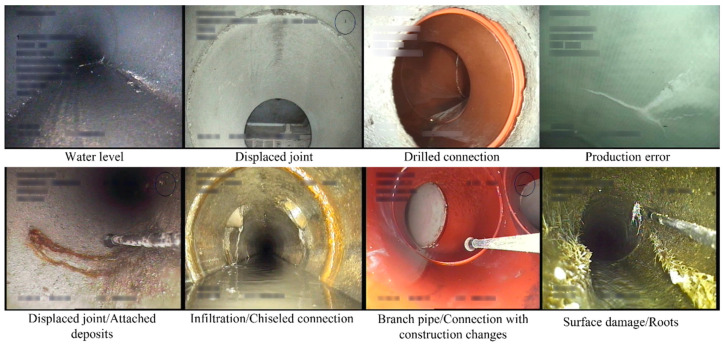
Sample images from the Sewer-ML dataset that has a wide diversity of materials and shapes.

**Figure 14 sensors-22-02722-f014:**
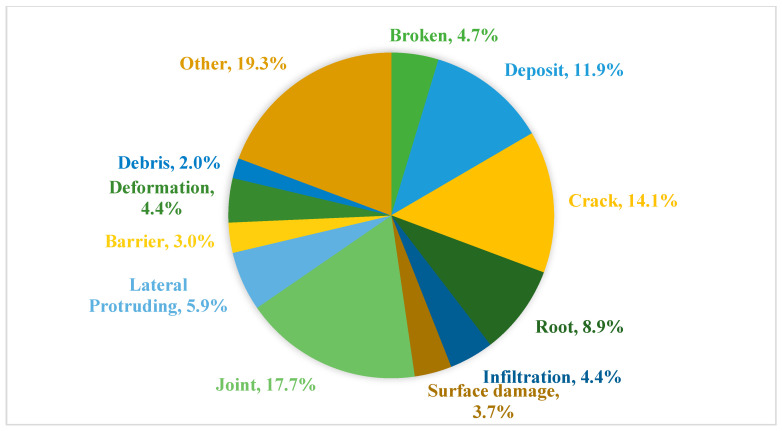
Studies on sewer inspections of different classes of defects.

**Table 1 sensors-22-02722-t001:** Major contributions of the previous review papers on defect inspection and condition assessment. ‘√’ indicates the research areas (defect inspection or condition assessment) are involved. ‘×’ means the research areas (defect inspection or condition assessment) are not involved.

ID	Ref.	Time	Defect Inspection	Condition Assessment	Contributions
1	[24]	2019	√	√	Analyze the status of practical defect detection and condition assessment technologies. Compare the benefits and drawbacks of the reviewed work.
2	[10]	2020	√	×	Introduce defect inspection methods that are suitable for different materials. Provide a taxonomy of various defects. List the state-of-the-art (SOTA) methods for the classification and detection.
3	[7]	2020	√	×	Create a brief overview of the defect inspection algorithms, datasets, and evaluation metrics. Indicate three recommendations for the future research.
4	[20]	2020	×	√	Investigate different models for the condition assessment. Analyze the influence factors of the reviewed models on the sewer conditions.
5	[8]	2021	√	√	Present a review for main applications, advantages, and possible research areas.

**Table 5 sensors-22-02722-t005:** The detailed information of the latest robots for sewer inspection.

Name	Company	Pipe Diameter	Camera Feature	Country	Strong Point
CAM160 (https://goolnk.com/YrYQob accessed on 20 February 2022)	Sewer Robotics	200–500 mm	NA	USA	● Auto horizon adjustment ● Intensity adjustable LED lighting ● Multifunctional
LETS 6.0 (https://ariesindustries.com/products/ accessed on 20 February 2022)	ARIES INDUSTRIES	150 mm or larger	Self-leveling lateral camera or a Pan and tilt camera	USA	● Slim tractor profile ● Superior lateral camera ● Simultaneously acquire mainline and lateral videos
wolverine® 2.02	ARIES INDUSTRIES	150–450 mm	NA	USA	● Powerful crawler to maneuver obstacles ● Minimum set uptime ● Camera with lens cleaning technique
X5-HS (https://goolnk.com/Rym02W accessed on 20 February 2022)	EASY-SIGHT	300–3000 mm	≥2 million pixels	China	● High-definition ● Freely choose wireless and wired connection and control ● Display and save videos in real time
Robocam 6 (https://goolnk.com/43pdGA accessed on 20 February 2022)	TAP Electronics	600 mm or more	Sony 130-megapixel Exmor 1/3-inch CMOS	Korea	● High-resolution ● All-in-one subtitle system
RoboCam Innovation4	TAP Electronics	600 mm or more	Sony 130-megapixel Exmor 1/3-inch CMOS	Korea	● Best digital record performance ● Super white LED lighting ● Cableless
Robocam 30004	TAP Electronics’ Japanese subsidiary	250–3000 mm	Sony 1.3-megapixel Exmor CMOS color	Japan	● Can be utilized in huge pipelines ● Optical 10X zoom performance

**Table 6 sensors-22-02722-t006:** Research datasets for sewer defects in recent studies.

ID	Defect Type	Image Resolution	Equipment	Number of Images	Country	Ref.
1	Broken, crack, deposit, fracture, hole, root, tap	NA	NA	4056	Canada	[9]
2	Connection, crack, debris, deposit, infiltration, material change, normal, root	1440 × 720–320 × 256	RedZone^®^ Solo CCTV crawler	12,000	USA	[48]
3	Attached deposit, defective connection, displaced joint, fissure, infiltration, ingress, intruding connection, porous, root, sealing, settled deposit, surface	1040 × 1040	Front-facing and back-facing camera with a 185∘ wide lens	2,202,582	The Netherlands	[49]
4	Dataset 1: defective, normal	NA	NA	40,000	China	[69]
	Dataset 2: barrier, deposit, disjunction, fracture, stagger, water			15,000		
5	Broken, deformation, deposit, other, joint offset, normal, obstacle, water	1435 × 1054–296 × 166	NA	18,333	China	[70]
6	Attached deposits, collapse, deformation, displaced joint, infiltration, joint damage, settled deposit	NA	NA	1045	China	[41]
7	Circumferential crack, longitudinal crack, multiple crack	320 × 240	NA	335	USA	[11]
8	Debris, joint faulty, joint open, longitudinal, protruding, surface	NA	Robo Cam 6 with a 1/3-in. SONY Exmor CMOS camera	48,274	South Korea	[71]
9	Broken, crack, debris, joint faulty, joint open, normal, protruding, surface	1280 × 720	Robo Cam 6 with a megapixel Exmor CMOS sensor	115,170	South Korea	[52]
10	Crack, deposit, else, infiltration, joint, root, surface	NA	Remote cameras	2424	UK	[66]
11	Broken, crack, deposit, fracture, hole, root, tap	NA	NA	1451	Canada	[104]
12	Crack, deposit, infiltration, root	1440 × 720–320 × 256	RedZone^®^ Solo CCTV crawler	3000	USA	[98]
13	Connection, fracture, root	1507 × 720–720 × 576	Front facing CCTV cameras	3600	USA	[99]
14	Crack, deposit, root	928 × 576–352 × 256	NA	3000	USA	[97]
15	Crack, deposit, root	512 × 256	NA	1880	USA	[116]
16	Crack, infiltration, joint, protruding	1073 × 749–296 × 237	NA	1106	China	[122]
17	Crack, non-crack	64 × 64	NA	40,810	Australia	[109]
18	Crack, normal, spalling	4000 × 46,000–3168 × 4752	Canon EOS. Tripods and stabilizers	294	China	[73]
19	Collapse, crack, root	NA	SSET system	239	USA	[61]
20	Clean pipe, collapsed pipe, eroded joint, eroded lateral, misaligned joint, perfect joint, perfect lateral	NA	SSET system	500	USA	[56]
21	Cracks, joint, reduction, spalling	512 × 512	CCTV or Aqua Zoom camera	1096	Canada	[54]
22	Defective, normal	NA	CCTV (Fisheye)	192	USA	[57]
23	Deposits, normal, root	1507 × 720–720 × 576	Front-facing CCTV cameras	3800	USA	[72]
24	Crack, non-crack	240 × 320	CCTV	200	South Korea	[106]
25	Faulty, normal	NA	CCTV	8000	UK	[65]
26	Blur, deposition, intrusion, obstacle	NA	CCTV	12,000	NA	[67]
27	Crack, deposit, displaced joint, ovality	NA	CCTV (Fisheye)	32	Qatar	[103]
29	Crack, non-crack	320 × 240–20 × 20	CCTV	100	NA	[100]
30	Barrier, deposition, distortion, fraction, inserted	600 × 480	CCTV and quick-view (QV) cameras	10,000	China	[110]
31	Fracture	NA	CCTV	2100	USA	[105]
32	Broken, crack, fracture, joint open	NA	CCTV	291	China	[59]

**Table 7 sensors-22-02722-t007:** Overview of the evaluation metrics in the recent studies.

Metric	Description	Ref.
Precision	The proportion of positive samples in all positive prediction samples	[9]
Recall	The proportion of positive prediction samples in all positive samples	[48]
Accuracy	The proportion of correct prediction in all prediction samples	[48]
F1-score	Harmonic mean of precision and recall	[69]
FAR	False alarm rate in all prediction samples	[57]
True accuracy	The proportion of all predictions excluding the missed defective images among the entire actual images	[58]
AUROC	Area under the receiver operator characteristic (ROC) curve	[49]
AUPR	Area under the precision-recall curve	[49]
mAP	mAP first calculates the average precision values for different recall values for one class, and then takes the average of all classes	[9]
Detection rate	The ratio of the number of the detected defects to total number of defects	[106]
Error rate	The ratio of the number of mistakenly detected defects to the number of non-defects	[106]
PA	Pixel accuracy calculating the overall accuracy of all pixels in the image	[116]
mPA	The average of pixel accuracy for all categories	[116]
mIoU	The ratio of intersection and union between predictions and GTs	[116]
fwIoU	Frequency-weighted IoU measuring the mean IoU value weighing the pixel frequency of each class	[116]

**Table 8 sensors-22-02722-t008:** Performances of different algorithms in terms of different evaluation metrics.

ID	Number of Images	Algorithm	Task	Performance		Ref.
				Accuracy (%)	Processing Speed	
1	3 classes	Multiple binary CNNs	Classification	Accuracy: 86.2 Precision: 87.7 Recall: 90.6	NA	[48]
2	12 classes	Single CNN	Classification	AUROC: 87.1 AUPR: 6.8	NA	[48]
3	Dataset 1: 2 classes	Two-level hierarchical CNNs	Classification	Accuracy: 94.5 Precision: 96.84 Recall: 92 F1-score: 94.36	1.109 h for 200 videos	[69]
	Dataset 2: 6 classes			Accuracy: 94.96 Precision: 85.13 Recall: 84.61 F1-score: 84.86		
4	8 classes	Deep CNN	Classification	Accuracy: 64.8	NA	[70]
5	6 classes	CNN	Classification	Accuracy: 96.58	NA	[71]
6	8 classes	CNN	Classification	Accuracy: 97.6	0.15 s/image	[52]
7	7 classes	Multi-class random forest	Classification	Accuracy: 71	25 FPS	[66]
8	7 classes	SVM	Classification	Accuracy: 84.1	NA	[41]
9	3 classes	SVM	Classification	Recall: 90.3 Precision: 90.3	10 FPS	[11]
10	3 classes	CNN	Classification	Accuracy: 96.7 Precision: 99.8 Recall: 93.6 F1-score: 96.6	15 min 30 images	[73]
11	3 classes	RotBoost and statistical feature vector	Classification	Accuracy: 89.96	1.5 s/image	[61]
12	7 classes	Neuro-fuzzy classifier	Classification	Accuracy: 91.36	NA	[56]
13	4 classes	Multi-layer perceptions	Classification	Accuracy: 98.2	NA	[54]
14	2 classes	Rule-based classifier	Classification	Accuracy: 87 FAR: 18 Recall: 89	NA	[57]
15	2 classes	OCSVM	Classification	Accuracy: 75	NA	[65]
16	4 classes	CNN	Classification	Recall: 88 Precision: 84 Accuracy: 85	NA	[67]
17	2 class	Rule-based classifier	Classification	Accuracy: 84 FAR: 21 True accuracy: 95	NA	[58]
18	4 classes	RBN	Classification	Accuracy: 95	NA	[59]
19	7 classes	YOLOv3	Detection	mAP: 85.37	33 FPS	[9]
20	4 classes	Faster R-CNN	Detection	mAP: 83	9 FPS	[98]
21	3 classes	Faster R-CNN	Detection	mAP: 77	110 ms/image	[99]
22	3 classes	Faster R-CNN	Detection	Precision: 88.99 Recall: 87.96 F1-score: 88.21	110 ms/image	[97]
23	2 classes	CNN	Detection	Accuracy: 96 Precision: 90	0.2782 s/image	[109]
24	3 classes	Faster R-CNN	Detection	mAP: 71.8	110 ms/image	[105]
		SSD		mAP: 69.5	57 ms/image	
		YOLOv3		mAP: 53	33 ms/image	
25	2 classes	Rule-based detector	Detection	Detection rate: 89.2 Error rate: 4.44	1 FPS	[106]
26	2 classes	GA and CNN	Detection	Detection rate: 92.3	NA	[100]
27	5 classes	SRPN	Detection	mAP: 50.8 Recall: 82.4	153 ms/image	[110]
28	1 class	CNN and YOLOv3	Detection	AP: 71	65 ms/image	[108]
29	3 classes	DilaSeg-CRF	Segmentation	PA: 98.69 mPA: 91.57 mIoU: 84.85 fwIoU: 97.47	107 ms/image	[116]
30	4 classes	PipeUNet	Segmentation	mIoU: 76.37	32 FPS	[122]

## Data Availability

Not applicable.

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
