# Peer review of "Vision-Based Defect Inspection and Condition Assessment for Sewer Pipes: A Comprehensive Survey"

_sensors, 2022, doi:10.3390/s22072722_

Round 1
Reviewer 1 Report
Very interesting work. A few remarks:
- In the abstract, after 'By reviewing...', I suggest anticipating in a few words the most salient outcomes of the paper
- Section 2: you should be more precise on indicating the keywords used in your bibliographic search to produce the outcomes included in Figure 2. For instance: "KEYWORD1" AND "KEYWORD2" OR "KEYWORD3".
- Better define the scope of the paper. State which construction materials that your survey looks at (plastic pipes, concrete pipes), as well as what the typical dimensions of the pipes are, etc. Every typology has specific problems (surface roughness and regularity, presence of fouling, etc. etc.). If the survey is general and not focused on some distinct materials and pipe typologies, please state it.
- Following the remark above, I suggest extending the scope of the paper. You mainly refer to concrete pipes in your survey. However, most of the European cities still make use of old aqueducts made of bricks vaults as part of the principal sewage system. The assessment of those structures is especially cumbersome for us in the practice because during the inspection the brickworks creates many false positives. Are there special automated defect-detection and survey techniques that can be used in those cases? It may be relevant for practitioners to include this aspect in the discussion. There seem to be a few works published on the topic, but they focus on structures that are outside the ground (10.1016/j.autcon.2021.103606, 10.1016/j.conbuildmat.2021.124831).
- The conclusions repeat what already contained in the abstract. It would be more interesting to use the conclusions to summarize the main outcomes of the survey (without repeating what contained in Section 5 though).
Reviewer 2 Report
The paper brings a comprehensive review of vision-based methods for sewer pipe inspection, with a focus on image processing algorithms. Overall, the study is very comprehensive and informative. The technical depth is satisfactory for a review paper.
I only have to minor comments:
- Please provide a clear explanation on what is the exact problem and where are the differences when it comes to object classification, detection and segmentation in a context of visual sewer inspection.
For instance, in Section 3.2. the authors state that "object detection ... is conducted to locate and classify the object among the predefined classes...". In Section 3.3. they state that "defect segmentation algorithms can predict defect categories and the precise location information...". Both definitions seem essentially the same?
2. The language and writing style should be fairly improved for better readability.
Reviewer 3 Report
A comprehensive survey of Vision-based defect inspection and condition assessment for sewer pipes was presented in the paper.
- The paper represent a solid research dedicated to assessment of effectiveness of vision-based inspection of sewer pipes.
- Even though vision-based method is probably the most cost effective method, it is missing the information about pipe OD surface condition. This is why it would be nice to see some short analysis of alternative methods such as ultrasonic wall thickness measurements, EMAT guided waves and their potential value for more comprehensive assessment of sewer pipe condition. I noticed that authors mentioned multi-sonar system in the reference 4. However, the method of anomaly detection and classification using sonars might be different from what is used in visual camera systems.
- Other than that I think it is a very valuable research paper.
Round 2
Reviewer 1 Report
The authors have carried out a good revision work.